# In Vitro and In Vivo Effects of Nobiletin on DRG Neurite Elongation and Axon Growth after Sciatic Nerve Injury

**DOI:** 10.3390/ijerph18178988

**Published:** 2021-08-26

**Authors:** Tae-Beom Seo, Yoon-A Jeon, Sang Suk Kim, Young Jae Lee

**Affiliations:** 1Department of Kinesiology, College of Natural Science, Jeju National University, Jeju 63243, Korea; seotb@jejunu.ac.kr; 2Laboratory of Veterinary Toxicology, College of Veterinary Medicine, Jeju National University, Jeju 63243, Korea; yoonaj0915@jejunu.ac.kr; 3Veterinary Medical Research Institute, Jeju National University, Jeju 63243, Korea; 4Citrus Research Institute, National Institute of Horticultural & Herbal Science, RDA, Seogwipo 63607, Korea; sskim0626@korea.kr

**Keywords:** sciatic nerve, nobiletin, extract, primary cell culture, dorsal root ganglion, in vivo, in vitro

## Abstract

Sciatic nerve injury (SNI) leads to sensory and motor dysfunctions. Nobiletin is a major component of polymethoxylated flavonoid extracted from citrus fruits. The role of nobiletin on sciatic nerve regeneration is still unclear. Thus, the purpose of this study was to investigate whether nobiletin increases DRG neurite elongation and regeneration-related protein expression after SNI. Cytotoxicity of nobiletin was measured in a concentration–dependent manner using the MTT assay. For an in vitro primary cell culture, the sciatic nerve on the middle thigh was crushed by holding twice with forceps. Dorsal root ganglion (DRG) and Schwann cells were cultured 3 days after SNI and harvested 36 h later and 3 days later, respectively. In order to evaluate specific regeneration-related markers and axon growth in the injured sciatic nerve, we applied immunofluorescence staining and Western blot techniques. Nobiletin increased cell viability in human neuroblastoma cells and inhibited cytotoxicity induced by exposure to H_2_O_2_. Mean neurite length of DRG neurons was significantly increased in the nobiletin group at a dose of 50 and 100 μM compared to those at other concentrations. GAP-43, a specific marker for axonal regeneration, was enhanced in injury preconditioned Schwann cells with nobiletin treatment and nobiletin significantly upregulated it in injured sciatic nerve at only 3 days post crush (dpc). In addition, nobiletin dramatically facilitated axonal regrowth via activation of the BDNF-ERK1/2 and AKT pathways. These results should provide evidence to distinguish more accurately the biochemical mechanisms regarding nobiletin-activated sciatic nerve regeneration.

## 1. Introduction

The peripheral nervous system (PNS) consists of somatic and autonomic nerves, and serves to transmit electrical impulses from the central nervous system (CNS) to peripheral tissues. The sciatic nerve, the largest peripheral nerve in the human body, has a high incidence of damage in sports or daily life due to various anatomical changes such as herniation of intervertebral discs or compression of the piriform muscle [1,2].

Sciatic nerve injury (SNI) has been known to induce sensory and motor dysfunctions as well as autonomic function disorders [3,4], and patients with SNI eventually suffer from long-term morbidity and incur high treatment costs. Previous studies on sciatic nerve regeneration suggested that Wallerian degeneration, in which axons in the distal segment disappear immediately after SNI, is the cause of these morbidity and economics [5]. Specifically, Wallerian degeneration after SNI leads to breakdown of Schwann cells around the injury area and into the distal nerve segment, and then the surviving Schwann cells secrete not only growth-associated protein-43 (GAP-43) that is a biochemical marker for axonal regeneration, but brain-derived neurotrophic factor (BDNF) and nerve growth factor (NGF) to induce Schwann cell proliferation and axon growth [6]. Our previous studies investigated showed that several proteins related with proliferation of Schwann cell including GAP-43 and neurotrophic factors dramatically increased at the initial stage of injury and then decreased from 7 days after SNI [7,8]. We thought that these findings were because the proliferation of Schwann cells occurs only in the early stage after SNI. Finally, Schwann cells promote remyelination-related myelin basic protein (MBP) expression in demyelinated sciatic axons [7,8]. In other words, SNI can be voluntarily treated, but there are temporal and economic difficulties in inducing complete functional recovery.

Nobiletin is a major nonpeptide component of polymethoxylated flavonoid extracted from citrus fruits and has several beneficial properties, such as regulating anti-inflammation in osteoarthritis, insulin resistance in obese diabetes and apoptosis in cancer cells [9,10,11]. In addition to these effects, nobiletin produces neurotrophic action in the degenerative nervous system [12,13]. Looking at previous studies reporting the effect of nobiletin in age-related degenerative diseases, nobiletin may be a novel compound for improving cognitive and memory deterioration in Alzheimer’s disease [14], and it has the therapeutic potential to treat neurodegenerative diseases through suppressing nuclear NF-kB translocation from the cytosol [15].

With these previous results, nobiletin is believed to be a substance of high interest for the treatment of degenerative and metabolic diseases. However, the molecular biological role of nobiletin at the early stage of peripheral nerve regeneration is still not clear. Therefore, the purpose of this study was to investigate in vitro and in vivo effects of nobiletin on neurite elongation of DRG neurons and regeneration-related protein expression, as well as axon growth after SNI.

## 2. Materials and Methods

### 2.1. Experimental Animals and Sciatic Nerve Injury

Male Sprague-Dawley rats (6 weeks old) were purchased from the Experimental Animal Center of Jeju National University (Jeju, Korea). The use of rats in this study was approved by Ethical committee of Jeju National University (approval number: 2020-0020). Experimental procedures were conducted in accordance with the guidelines for the Care and Use of Laboratory Animals at Jeju National University. Nobiletin was purchased from Sigma-Aldrich Co. (purity 97.0%; molecular weight 402.39; St. Louis, MO, USA). Nobiletin was obtained from National Institute of horticultural & Herbal Science in Korea (purity 98.0%; molecular weight, 402.93 g/mol) The animals for the in vivo experiment were divided into five groups with a randomization method: the normal group, SNI+vehicle at 1- and 3-days post crush (dpc) group, and SNI+nobiletin treated at 1 and 3 dpc group. All experiment animals used anesthetized using an animal inhalation narcosis control (Jeungdo bio & plant; Seoul, Korea). First, the rats were placed into a chamber with a 2–2.5% concentration of isoflurane for anesthesia, and then a 1.5–1.8% concentration for maintenance during sciatic nerve injury. The sciatic nerve was exposed on the middle thigh and crushed by holding twice with forceps for 1 min and 30 secs at intervals [7]. For examining the in vivo effect of nobiletin, 5 µL of Nobiletin at 50 µM concentration were injected into a point 10 mm distal to the injury site immediately after SNI using a micro-syringe. A vehicle solution (0.1% DMSO in 0.9% normal saline) was injected at the same location as the nobiletin treated group. The animals after surgery rested for 1 h on a heating pad at 37 °C and then were returned to their home cages. Sciatic nerves and lumbar 4–5 dorsal root ganglions (L4-5 DRG) were dissected according to the experimental schedule. 

### 2.2. MTT Assay 

Human neuroblastoma cells (SH-SY5Y) were maintained at 37 °C in an incubator in DMEM with 10% heat-inactivated fetal bovine serum (Hi-FBS, Gibco Inc., Billings, MT, USA) and 1% Penicillin-Streptomycin (Gibco Inc., Billings, MT, USA). Cells were seeded at a density of 2 × 10^4^ into 96 well cell culture plates. Sixteen hours after incubation, nobiletin was treated in each well for 1 h and then stressed with 600 μM H_2_O_2_ for 23 h. Cell viability was determined by adding 0.2 mg/mL of 3-(4,5-dimethylthiazol-2-yl)-2,5-diphenyltetrazolium bromide (MTT, Sigma Aldrich, St. Louis, MA, USA) and cells were further incubated for 4 h at 37 °C in an incubator. Cells were centrifuged 2000 rpm for 5 min and then the media was removed. After 200 µL of DMSO was added to each well the plates were shaken for 1 h. After cells were solubilized, density was determined by an ELISA reader (Magellan, Tecan, Männedorf, Switzerland) at 570 nm. Cell viability was presented as a percentage of vehicle control.

### 2.3. Primary DRG Neuron and Schwann Cell Culture 

The dorsal root ganglion (DRG) is a cluster of neurons in the dorsal root of the spinal nerve, and sensory neurons of the sciatic nerve are normally located in DRG at lumbar 4–5 [16]. In our previous studies, we confirmed retrograde tracing of L4-5 DRG neurons after injection of DiI into the distal region to the damage site after SNI [8]. Primary DRG sensory neurons and Schwann cells were prepared from L4-5 DRG and the sciatic nerve, respectively, as described previously [7,8,17]. For in vitro experiment of the DRG sensory neuron, a total six SD rats were used and this experiment was repeated three times with two rats each. To observe GAP-43 expression levels in primary cultured Schwann cells, SD rats (*n* = 18) were assigned to normal control, vehicle and nobiletin treated groups (*n* = 2, each) with experiments repeated three times. Three days after SNI, the sciatic nerve and DRG at L4-5 were separated in DMEM for 60 min at 37 °C. Cells were treated with trypsin for 15 min and followed by inhibition reaction for 5 min in trypsin inhibitor. DRG neurons (1 × 10^5^ cells per dish) were plated onto 12 mm coverslips (Bellco Glass Inc., Vineland, NJ, USA) precoated with 0.01% poly-L-ornithine (Sigma Aldrich, St. Louis, MA, USA) and laminin (0.02 mg/mL) (Roche Diagnostics, Mannheim, Germany). Twelve hours after DRG neuron culture, DMEM was changed and nobiletin was added to each well at concentrations of 0, 1, 30, 50, 100 or 200 µM. A vehicle solution (0.1% DMSO + 0.1% EtOH in DMEM) was treated in a culture dish for the control group. Neurons were cultured for 36 h and harvested for immunofluorescence staining. For detection of regeneration-related proteins, Schwann cells (5 × 10^6^) were cultured in a 60 mm culture dish, and 12 h later nobiletin was added to each well at a concentration of 50 µM. Schwann cells were incubated for 72 h and then harvested for Western blot analysis.

### 2.4. Western Blot Analysis

For Western blot expression of GAP-43 and BDNF, ERK1/2 and AKT signaling pathways in the sciatic nerve, SD rats (*n* = 16) were assigned to SNI+vehicle at 1 and 3 dpc groups, and SNI+nobiletin treated at 1 and 3 dpc groups (*n* = 4, each). The nerve segments and Schwann cells were washed with ice-cold PBS and sonicated under 400 mL of Triton lysis buffer (20 mM Tris [pH 7.4], 137 mM NaCl, 25 mM β-glycerophosphate [pH 7.14], 2 mM sodium pyrophosphate, 2 mM EDTA, 1 mM Na_3_VO_4_, 1% Triton X-100, 10% glycerol, 5 mg/mL leupeptin, 5 mg/mL aprotinin, 3 mM benzamidine, 0.5 mM DTT, and 1 mM PMSF). The cell lysate from the nerve stump and primary cultured Schwann cells was prepared as described elsewhere [8]. Protein (15 µg) was used for Western blot analysis using antigrowth associated protein-43 (GAP-43) mouse monoclonal antibody (1:1000) (Santa Cruz Biotechnology, Dallas, TX, USA), antiphosphorylated ERK1/2 rabbit polyclonal antibody (1:2000) (Cell Signaling Biotechnology, Danvers, MA, USA), antiphosphorylated AKT rabbit monoclonal antibody (1:1000) (Cell Signaling Biotechnology, Danvers, MA, USA), anti-BDNF rabbit polyclonal antibody (1:1000) (Santa Cruz Biotechnology, Dallas, TX, USA), and goat anti-mouse or goat anti-rabbit horseradish peroxidase conjugated secondary antibody (1:1000) (GeneTex Inc., Irvine, CA, USA) were used. The blotting proteins were detected by using chemiluminescence (Westar sun; Cyanagen, Bologna, Italy). Analysis of protein density was performed using Chemidoc (Bio-Rad, Hercules, CA, USA).

### 2.5. Immunofluorescence Staining 

For detection of regenerating axons in the distal region of the injured sciatic nerves, SD rats (*n* = 10) were assigned to normal control (*n* = 2), vehicle and nobiletin treated groups (*n* = 4, each). Nerve segments were embedded and frozen at 20 °C. Longitudinal or transverse sections (20 µm thick) were cut on a cryostat and mounted on positively charged slides (Fisher Scientific, Pittsburgh, PA, USA). For immunofluorescence staining, sections and primary cultured DRG neurons were fixed with 4% paraformaldehyde and 4% sucrose in PBS at room temperature for 40 min, permeabilized with 0.5% Nonidet P-40 in PBS, and blocked with 2.5% horse serum and 2.5% bovine serum albumin for 4 h at room temperature. The sections were incubated with antineurofilament-200 (NF-200) rabbit polyclonal antibody (1:700) (Sigma-Aldrich, St. Louis, MA, USA) and Hoechst. Then they were incubated with rhodamine-goat anti-rabbit secondary antibody (1:600) (Molecular Probes, Eugene, OR, USA) for 1 h at room temperature. The stained samples were viewed with a fluorescence microscope (Nikon model E-600; Nikon, Japan), and the images were captured with a digital camera and analyzed using Adobe Photoshop Software (version CS6; Adobe, San Jose, CA, USA). The number and length of DRG neurites were evaluated by using i-Solution software (Image and Microscope Technology, Irvine, CA, USA). Mean neurite length of primary cultured DRG neurons was measured by analyzing at least 30 sensory neurons which were randomly selected from each experiment. The I of regenerating axons in the injured sciatic nerve was counted from three or four nonconsecutive sections. Analysis on the neurite length in DRG neurons and the number of regenerating axons was conducted by an examiner blinded to the experimental treatment conditions. 

### 2.6. Statistical Analysis

The PASW (Statistical Package for Predictive Analytics Soft Ware) 18.0 program was used to confirm differences between groups. The number and length of DRG neurites, cell viability, and GAP-43 levels in Schwann cells were performed using one-way ANOVA followed by the Duncan post hoc test. Proteins produced in the injured sciatic nerves were analyzed by independent *t*-tests. All data is presented as a mean ± standard error, and the significance level was set at *p* < 0.05. All graphs were produced using Prism 6 (GraphPad, San Diego, CA, USA).

## 3. Results

### 3.1. Nobiletin Suppresses the Loss of Cell Viability in SH-SY5Y Cells

To investigate the effect of nobiletin on cell viability, we analyzed the cytotoxicity of nobiletin in a concentration-dependent method using the SH-SY5Y cells. As shown in Figure 1a, cell population was significantly increased with 3.12 to 50 μM concentration of nobiletin, but there was no difference at low concentrations of nobiletin. The number of SH-SY5Y cells was dramatically decreased by exposure to H_2_O_2_, but treatment of nobiletin from low concentration to 50 μM significantly improved viability of SH-SY5Y cells (Figure 1b).

### 3.2. Nobiletin Dose-Dependently Regulates Neurite Outgrowth of DRG Neurons

To examine changes in neurite processes of primary cultured DRG neurons at 3 days after SNI with various concentration of nobiletin, we added 0, 1, 30, 50, 100 or 200 μM of nobiletin to each well. Neurite outgrowth of DRG neurons has been recognized as an indicator of indirect peripheral nerve axon regeneration [16]. As shown in Figure 2, treatment with nobiletin at a dose of 50 and 100 μM significantly increased mean neurite length of DRG neurons compared to those in other concentration. 

### 3.3. Nobiletin Increases GAP-43 Expression In Vitro and In Vivo

To examine GAP-43 expression in vitro and in vivo, we performed Western blot analysis using primary cultured Schwann cells and the injured nerves. As shown in Figure 3a, GAP-43 expression in injury-preconditioned Schwann cells was significantly enhanced in the nobiletin treated group compared to other groups. Moreover, in the injured sciatic nerves, GAP-43 was dramatically upregulated at 3dpc and showed a significant difference in the nobiletin-treated group at only 3dpc compared to the vehicle-treated group (Figure 3b).

### 3.4. Nobiletin Controls Activation of BDNF, ERK1/2 and AKT Signaling Pathways at Early Stage of Nerve Regeneration

To examine the expression of regeneration-related proteins, we analyzed time-dependent alterations in BDNF, ERK1/2 and AKT at an early stage of nerve regeneration. As shown in Figure 4, p-ERK1/2 in the nobiletin group was significantly increased at only 3 dpc compared to the vehicle group, but BDNF in the nobiletin group showed a tendency to upregulate from 1 to 3 dpc. Moreover, p-AKT levels in the vehicle group did not change over time after SNI, whereas nobiletin treatment significantly increased p-AKT expression until 3 dpc. 

### 3.5. Nobiletin Facilitated Axonal Regrowth at 2 Weeks after SNI 

To confirm axonal elongation in the injured sciatic nerve, sciatic nerves were prepared 2 weeks after SNI. Immunofluorescence images of NF-200-labeled axons showed that nobiletin further facilitated axonal elongation in the stump 5 and 10 mm distal to the crush site 2 weeks after SNI compared to those in DMSO group site (Figure 5).

## 4. Discussion

Recent studies on effect of citrus flavonoids in animal and clinical experiments have shown that nobiletin can mediate cell proliferation, differentiation and inflammation in various diseases including metabolic diseases, cancer and age-related degenerative diseases [18,19,20]. However, these studies have limitations in which research on peripheral nerve injury that occurs easily in everyday life have been excluded. Thus, we tried to demonstrate the effect of nobiletin on sciatic nerve regeneration.

Nobiletin is a dietary polymethoxylated flavonoid extracted from citrus peels and it has few side effects on the human body [21]. Our study first confirmed the effect of nobiletin on cell survival. Nobiletin increased the number of SH-SY5Y cells, and treatment of nobiletin at low concentration to 50 μM significantly improved cell viability reduced by exposure to H_2_O_2_. In previous studies focused on the protective effect of nobiletin, Lu et al. [22] reported that the citrus flavonoid nobiletin suppressed cytotoxicity of PC12 cells induced by H_2_O_2_ and is one of various candidates for improving neurodegenerative diseases. Cho et al. [23] also noted that nobiletin might protect against hydrogen peroxide-induced cell death in HT22 hippocampal neuronal cells through regulating migogen-activated protein kinases and apoptotic pathways [24,25,26]. The results of these previous studies support our findings that nobiletin is a potential nontoxic material to activate the survival and function of neurons.

In the field of peripheral nerve regeneration, neurite outgrowth of DRG neurons (DRGs) is a main indicator to predict peripheral axonal regrowth after injury. de Siqueira-Santos et al. [27] suggested that extending sensory fibers of DRGs in vitro might be a regenerative property for promoting functional recovery after SNI, and Bucan et al. [28] reported that DRG neurite elongation activated by exosomes forms adipose derived mesenchymal stem cells closely involved in peripheral nerve regeneration. In the present study, nobiletin dose-dependently regulated neurite outgrowth of primary cultured DRGs after SNI and DRG neurite elongation was significantly facilitated at 50 and 100 μM of nobiletin compared to other dosages. According to previous studies on in vitro and in vivo effects of flavonoids on neuropathic pain, several flavonoids, including nobiletin, can attenuate acute and chronic peripheral neuropathic pain through regulating biochemical and biological levels in DRGs of animal models [3]. These data indicate that nobiletin might be a therapeutic compound for anterograde transport of regeneration-related signals from DRGs and suppression of neuropathic pain through stimulation of DRGs after SNI.

Our study investigated GAP-43 expression and axon growth in vitro and in vivo after SNI. Specifically, nobiletin significantly increased GAP-43 levels in injury-preconditioned Schwann cells and injured sciatic nerves until 3 dpc, as well as enhanced NF-200-stained regenerating axons in the region distal to the injury site 2 weeks after crushing. GAP-43 is an anterograde and/or retrograde transport protein controlling axonal regeneration in damaged peripheral nerves [29], and is localized to the axonal growth cone to promote neurite branching and synaptogenesis post injury [30]. Seo et al. [7] confirmed colocalization of GAP-43 and regenerating axons at the early stage of sciatic nerve regeneration. Based on the relationship between GAP-43 and axonal regrowth in the nobiletin-treated group following crushing, we speculate that nobiletin contributed to the initiation of the peripheral nerve regeneration program.

In addition to examination of functions of GAP-43, we confirmed that BDNF induced by nobiletin was expressed in a similar pattern to GAP-43 in the injured nerves. BDNF has been known as a regulator to enhance GAP-43 in the injured DRGs and Schwann cells [30,31]. Neurotrophic factors, including BDNF and NGF, are also main upstream molecules to stimulate ERK1/2 and Akt phosphorylation [14,32]. In previous studies on regenerative mechanisms of damaged peripheral nerves, activation of the BDNF-ERK1/2 signaling pathway reduced neuropathic pain and enhanced axonal regeneration after SNI, increased Akt phosphorylation in DRGs regulating neurite branching [7], with co-activation of ERK1/2 and Akt leading to promotion of neurite elongation and neuronal survival in primary DRG culture [33,34]. In the present study, nobiletin dramatically activated p-ERK1/2 and Akt with increasing BDNF in the injured sciatic tissues at 3 dpc, suggesting that nobiletin might be an agent that facilitates axonal regrowth via activation of BDNF-ERK1/2 and AKT pathways.

Given the findings reported in previous and present studies, citrus flavonoids such as nobiletin may have important roles to facilitate sciatic nerve regeneration in vitro and in vivo.

## 5. Conclusions

We investigated whether nobiletin, a flavonoid extracted in citrus fruits, could activate DRG axon elongation, regeneration-related protein expression and axonal regrowth after SNI by applying in vivo and in vitro experimental methods. The present findings should provide evidence to distinguish more accurately the biochemical mechanisms regarding nobiletin-activated sciatic nerve regeneration.

## Figures and Tables

**Figure 1 ijerph-18-08988-f001:**
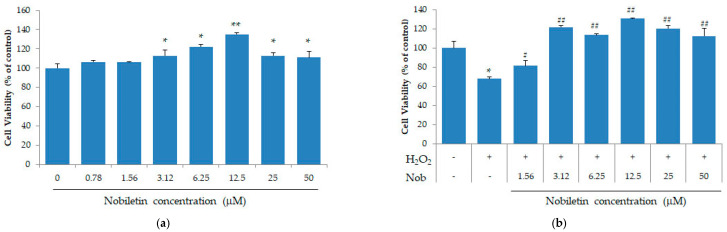
Nobiletin increased cell viability in human neuroblastoma cells and inhibited cytotoxicity induced by exposure to H_2_O_2_. (**a**) SH-SY5Y cells were cultured for 16 h and then nobiletin were added to each well. The differences of the cell population were determined by ELISA. Cell proliferation was significantly increased with 3.12 to 50 μM concentrations of nobiletin. * *p* < 0.05, ** *p* < 0.01 compared to vehicle treated group; (**b**) For cytotoxicity of nobiletin, SH-SY5Y cells were incubated into DMEM containing H_2_O_2_. Viability of H_2_O_2_-treated cells was significantly enhanced with 1.56 to 50 μM concentrations of nobiletin. * *p* < 0.05 compared to no treated group. # *p* < 0.05, ## *p* < 0.01 compared to only H_2_O_2_ treated group. Nob, nobiletin.

**Figure 2 ijerph-18-08988-f002:**
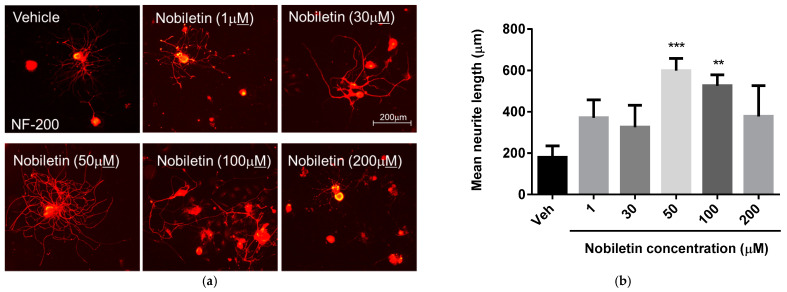
Nobiletin enhanced neurite outgrowth of primary cultured DRG neurons. For the effect of nobiletin on neuronal cell morphology, L4-5 DRG neurons were dissociated 3 days after SNI and plated on 60 mm culture dish. Twelve hours later nobiletin was treated in a concentration–dependent manner. Following 36 h, cultured DRG neurons were stained with anti-NF-200 antibody. (**a**) Representative images of neurite outgrowth of DRG sensory neurons after exposure to different concentration of nobiletin in the culture dishes. (**b**) Quantitative graph of DRG neurite length. Nobiletin at a dose of 50 and 100 μM significantly increased mean neurite length of DRG neurons (*n* = 6, this culture repeated three times with two rats each). ** *p* < 0.01, *** *p* < 0.001 compared to vehicle treated group; Veh, vehicle group, NF-200, neurofilamen-200.

**Figure 3 ijerph-18-08988-f003:**
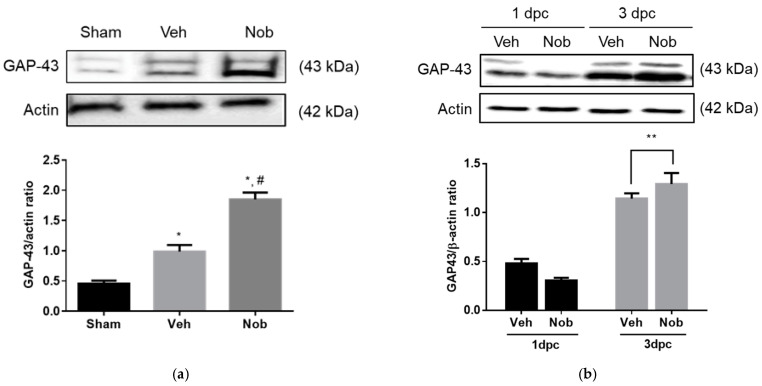
Nobiletin upregulated GAP-43 levels both in primary cultured Schwann cells and in the injured sciatic nerve. (**a**) Schwann cells were cultured at 3 dpc, and 12 h later, nobiletin was added to each well at a concentration of 50 µM. Schwann cells were incubated for 72 h and then harvested for Western blot analysis. Nobiletin upregulated expression levels of GAP-43 in injury-preconditioned Schwann cells compared to the vehicle group (*n* = 18, this culture repeated 3 times with 2 rats each). * *p* < 0.05 compared to shame group; # *p* < 0.05 compared to vehicle treated group. (**b**) Nobiletin at a concentration of 50 µM was locally injected into a segment 10 mm distal to the injury site, and 1 and 3 days later all sciatic nerves were dissected. The quantitative graph represented that GAP-43 showed a significant increase in the nobiletin treated group at only 3 dpc compared to subjects in the vehicle treated group (*n* = 16). ** *p* < 0.01 compared to vehicle treated group; Veh, vehicle group, Nob, nobiletin group; dpc, days post crush.

**Figure 4 ijerph-18-08988-f004:**
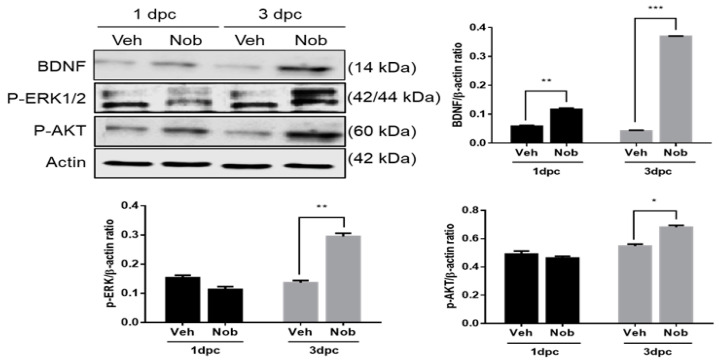
Nobiletin activated BDNF, ERK1/2 and AKT signaling pathways at 3 dpc and facilitated axon growth 2 weeks after SNI. (**a**) At the initial stage of sciatic nerve regeneration, nobiletin enhanced BDNF induction levels in the injured tissues at both 1 and 3 dpc, and ERK1/2 and AKT were significantly activated by nobiletin treatment at only 3 dpc. * *p* < 0.05, ** *p* < 0.01, *** *p* < 0.001 compared to the vehicle treated group.

**Figure 5 ijerph-18-08988-f005:**
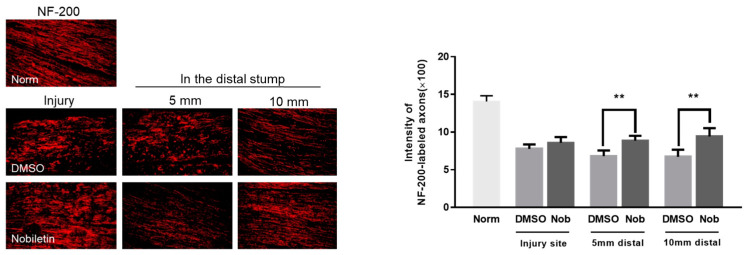
Nobiletin increased axonal elongation 2 weeks after SNI. Sciatic nerves were used for immunofluorescence staining with anti-NF-200 antibody. The nobiletin treated group showed apparently stronger staining of NF-200 in the segment 5 and 10 mm distal to the injury site than DMSO treated group. In each experiment, the intensity of axons stained with NF-200 was measured at injury site, 3 and 5 mm distal to the damage site from three or four nonconsecutive sections. There was a significant difference between DMSO- and nobiletin-treated groups (*n* = 10, this experiment were assigned to normal control (*n* = 2), vehicle and nobiletin-treated groups (*n* = 4, each). Norm, normal group; NF-200, neurofilamen-200. ** *p* < 0.01 compared to the DMSO-treated group.

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
