# Peer review of "In Vitro and In Vivo Effects of Nobiletin on DRG Neurite Elongation and Axon Growth after Sciatic Nerve Injury"

_ijerph, 2021, doi:10.3390/ijerph18178988_

Round 1

Reviewer 1 Report

Seo and colleagues studied the interesting topic of in vitro and in vivo effects of nobiletin on the injured sciatic nerve. They evaluated specific regeneration-related 18 markers and axon growth in the injured sciatic nerve and they found that nobiletin enhanced GAP-43, a specific marker for axonal regeneration and facilitates axonal regrowth via activation 25 of BDNF-ERK1/2 and AKT pathways.

However, there are several minor issues that have to address:

- The Material and Methods part lacks details. How many samples per group have been used in each experiment? Please provide relevant information in detail in method and figure legends.

- in Fig. 2, How many slices per animal have been used in the IHC? Distance between slices? The quality of immunohistochemical images is poor. Please replace it with a higher resolution image.

- Please provide the molecular weight markers in western blot data in fig. 3 & 4. Especially, BDNF is synthesized as a 32 kD pro-form which is proteolytically cleaved to the 14 kD mature form (mBDNF). The diverse biological functions of pro-form and mBDNF are mediated through a dual-receptor system, consisting of the tyrosine kinase receptor B (trkB) and pan-neurotrophin receptor p75. Therefore, the author needs to provide detailed information on what form of BDNF is and the molecular weight markers.

Author Response

  1. The Material and Methods part lacks details. How many samples per group have been used in each experiment? Please provide relevant information in detail in method and figure legends.

----- As pointed out by the reviewer, we added a detailed explanation of the method and figure legends. Please check ‘Materials and Methods’ and ‘figure legends’

  1. in Fig. 2, How many slices per animal have been used in the IHC? Distance between slices?

------ The quantitative analysis method requested by the reviewer was revised and improved as follows.

------ Mean neurite length of primary cultured DRG neurons was measured by analyzing at least 30 sensory neurons which were randomly selected from each experiment. And the I of regenerating axons in the injured sciatic nerve was counted from three or four non-consecution sections. Analysis on the neurite length in DRG neurons and the number of regenerating axons was conducted by an examiner blinded to the experimental treatment conditions.

------ For in vitro experiment of DRG sensory neuron, a total 6 SD rats were used and this experiment was repeated 3 times with 2 rats each. To observe GAP-43 expression levels in primary cultured Schwann cells, SD rat (n=18) were assigned to normal control, vehicle and nobiletin treated groups (n=2, each) and it repeated 3 times.

  1. Please provide the molecular weight markers in western blot data in fig. 3 & 4. Especially, BDNF is synthesized as a 32 kD pro-form which is proteolytically cleaved to the 14 kD mature form (mBDNF). The diverse biological functions of pro-form and mBDNF are mediated through a dual-receptor system, consisting of the tyrosine kinase receptor B (trkB) and pan-neurotrophin receptor p75. Therefore, the author needs to provide detailed information on what form of BDNF is and the molecular weight markers.

------ The molecular weight of all proteins analyzed by western blot are presented in the figure. Please check the results.  

Reviewer 2 Report

The author conducted aimed to investigate the effect of nobiletin extracted in citrus fruits, with the test on neuroblastoma cell line, primary culture DRG/Schwann cells, and crush sciatic nerve. Overall, the results are not coherent with each other, with lacking of some important evidence. The followings are the specific comments.

  1. In figure 1 for cytotoxicity test, what is the rationale to use human neuroblastoma cell line as representatives? Basically the SH-SY5Y cells are immortalized CNS dopaminergic cells with dopaminergic activities. Usually it is used to study Parkinson's disease, CNS neurogenesis, and other characteristics of brain cells. Therefore, the clinical translational significance to PNS neurogenesis is lacking.
  2. What is the rationale of locally treated Nobiletin distal to injury site? What is the target responsive cells the authors expect to take action. The author should provide the in vivo evidence how the local application of Nobiletin (distal to crush site) might affect the DRG neuron retrogradely, instead of in vitro effect on cultured DRG cells only.
  3. The only evidence provided by authors were mostly short-term effect. How did the locally applied Nobiletin improve the functional results, such sensory and motor recovery?
  4. After nerve crush, the nerve will undergo distal demyelination and Wallerian degeneration during early time period, followed by remyelination and regeneration. In figure 4a, the author provided regeneration associated neurotrophic cytokine within 3 days after injury, the longer result should be provided.
  5. For the control group (Veh), what exactly the author applied to the sciatic nerve? Did author exclude the placebo effect of the DMSO (solvent of Nobiletin)?
  6. The information of Nobiletin is lacking, including preparation, company, catalog number. If that is natural product, then the production process should be described in detail.
  7. For the results in most of the figures, author should provide N for both in vitro and in vivo test.
  8. Author should provide quantification, scale bar in figure 4b.

Author Response

  1. In figure 1 for cytotoxicity test, what is the rationale to use human neuroblastoma cell line as representatives? Basically the SH-SY5Y cells are immortalized CNS dopaminergic cells with dopaminergic activities. Usually it is used to study Parkinson's disease, CNS neurogenesis, and other characteristics of brain cells. Therefore, the clinical translational significance to PNS neurogenesis is lacking.

----- As pointed out by the reviewer, SH-SY5Y cells are a human derived cell used in vitro models of neuronal function and differentiation.

----- Regeneration of the sciatic nerve is closely related to the functional improvement of Schwann cells at the distal region of the injured nerves as well as the anterograde transport of regenerative proteins sent from motor neuron in the ventral horn of the spinal cord to the injury nerve.

----- In the present study, we want to know the toxicity of nobiletin in motor neurons, but it is known that the culture of primary motor neuron in the spinal cord is impossible.

----- For this reason, we investigated the toxicity of nobiletin using SH-SY5Y cells.

----- Thank you for your comments.

  1. What is the rationale of locally treated Nobiletin distal to injury site? What is the target responsive cells the authors expect to take action. The author should provide the in vivo evidence how the local application of Nobiletin (distal to crush site) might affect the DRG neuron retrogradely, instead of in vitro effect on cultured DRG cells only.

----- The reviewer’s comments were revised and improved as follows.

------ A dorsal root ganglion (DRG) is a cluster of neurons in a dorsal root of the spinal nerve, and sensory neurons of the sciatic nerve were normally located in DRG at lumbar 4-5 (Lyu et al., 2020). In our previous studies, we confirmed retrograde tracing of L4-5 DRG neurons after injection of DiI into the distal region to the damage site after SNI (Seo et al., 2009). Primary DRG sensory neurons and Schwann cells were prepared from L4-5 DRG and sciatic nerve, respectively, as described previously (Han et al., 2007; Seo et al., 2006, 2009).

3. The only evidence provided by authors were mostly short-term effect. How did the locally applied Nobiletin improve the functional results, such sensory and motor recovery?
-----The reviewer’s comments were revised and improved as follows.

------ The surviving Schwann cells secrete not only growth-associated protein-43 (GAP-43) that is a biochemical marker for axonal regeneration but also brain-derived neurotrophic factor (BDNF) and nerve growth factor (NGF) to induce Schwann cell proliferation and axon growth [6]. Our previous studies investigated that several proteins related with proliferation of Schwann cell including GAP-43 and neurotrophic factors dramatically increased at the initial stage of injury and then decreased from 7 days after SNI (Seo et al., 2006, 2009). We thought that these findings are because the proliferation of Schwann cells occurs only in the early stage after SNI.

4. After nerve crush, the nerve will undergo distal demyelination and Wallerian degeneration during early time period, followed by remyelination and regeneration. In figure 4a, the author provided regeneration associated neurotrophic cytokine within 3 days after injury, the longer result should be provided.

----- In general, it has been well known that Schwann cell proliferation induces immediately after peripheral nerve injury, and this phenomenon is the most important for axonal regeneration and functional recovery post SNI as well as proliferating Schwann cells secrets various growth factors to improve axonal growth and remyelination in the injured sciatic nerve. So, if Schwann cell proliferation may not be produced in the early stage of peripheral nerve regeneration, we cannot expect axonal regrowth and remyelination in the late stage of regeneration. Thus, in this manuscript, we suggested immunostaining results on NF-200-labeled regenerating axons 2 weeks after SNI. I don’t think the 2 weeks of regeneration period is the initial stage of regeneration after crush injury (not transection). Thank you for your comments.

5. For the control group (Veh), what exactly the author applied to the sciatic nerve? Did author exclude the placebo effect of the DMSO (solvent of Nobiletin)?
----- The reviewer’s comments were revised and improved as follows.

----- For examining in vivo effect of nobiletin, 5 µl of Nobiletin at 50 µM concentration were injected into the point 10 mm distal to the injury site immediately after SNI using micro-syringe. A vehicle solution (0.1% DMSO in 0.9% normal saline) was injected at the same location as the nobiletin treated group.

----- Twelve hours after DRG neuron culture, DMEM was changed and then nobiletin was treated into each well at a concentration of 0, 1, 30, 50, 100 or 200 µM. A vehicle solution (0.1% DMSO+0.1% EtOH in DMEM) was treated in culture dish of control group. Neurons were cultured for 36 hours and harvested for immunofluorescence staining. For detection of regeneration-related proteins, Schwann cells (5x106) were cultured onto 60 mm culture dish, and 12 hours later nobiletin was treated into each well at a concentration of 50 µM. Schwann cells were incubated for 72 h and then harvested for western blot analysis.

6. The information of Nobiletin is lacking, including preparation, company, catalog number. If that is natural product, then the production process should be described in detail.
----- The reviewer’s comments were revised and improved as follows.

----- Nobiletin was purchased from Simga-Aldrich Co. (purity 97.0%; molecular weight 402.39; St. Louis, MO, USA).

7. For the results in most of the figures, author should provide N for both in vitro and in vivo test.

----- The quantitative analysis method requested by the reviewer was revised and improved as follows.

----- For in vitro experiment of DRG sensory neuron, a total 6 SD rats were used and this experiment was repeated 3 times with 2 rats each. To observe GAP-43 expression levels in primary cultured Schwann cells, SD rat (n=18) were assigned to normal control, vehicle and nobiletin treated groups (n=2, each) and it repeated 3 times.

----- For detection of regenerating axons in the distal region of the injured sciatic nerves, SD rat (n=10) were assigned to normal control (n=2), vehicle and nobiletin treated groups (n=4, each).

8. Author should provide quantification, scale bar in figure 4b.

- Quantitative results and scale bar are added in figure 4b.

Reviewer 3 Report

The manuscript by Seo et al. deals with the use of a flavonoid called nobiletin in axonal recovery after crushing the sciatic nerve. The results point to an important contribution of the substance in the regenerative process by activation of BDNF produced by Schwann cells. The work is  interesting but needs some adjustments in the text as suggested below:

A review of the text is necessary as it contains typos or lack of commas. Some of these are:  lines 75, 88, 108, 144.

Material and methods:

1- The number of rats used in each technique?

2- Where are the controls? Since DMSO in high concentration is toxic and in low concentration it may have anti-inflammatory properties, it is important that you consider this and clearly state which concentration is used and its controls (item 2.1).

3- What is the concentration of Nobiletin used in vitro? Was nobiletin diluted in DMSO? What is the concentration of DMSO? If so, the control must have the same DMSO concentration (item 2.2).

4- Why did not you use one of the primary cultures (from uninjured tissue) to analyze the cytotoxicity of nobiletin? (item 2.2).

5- It is known that nerve and DRG have connective tissue envelopes. What was the procedure to ensure sufficient purity of the primary cultures ? (item 2.3).

6- How much protein was used in electrophoresis? The WB was performed in how many cultures? What is “N” in vitro and in vivo? Did you use tissue pool in the electrophoseris? (item 2.4).

Discussion:

“Zhang et al. [21] also noted that nobiletin inhibited proliferation and differentiation of tumor cells such as neuroblastoma, breast cancer and fibrosarcoma [22-24]”.

Where does this argument fit into the results?

Author Response

Material and methods:

  1. The number of rats used in each technique?

----- The quantitative analysis method requested by the reviewer was revised and improved as follows.

------ For in vitro experiment of DRG sensory neuron, a total 6 SD rats were used and this experiment was repeated 3 times with 2 rats each. To observe GAP-43 expression levels in primary cultured Schwann cells, SD rat (n=18) were assigned to normal control, vehicle and nobiletin treated groups (n=2, each) and it repeated 3 times.

----- For detection of regenerating axons in the distal region of the injured sciatic nerves, SD rat (n=10) were assigned to normal control (n=2), vehicle and nobiletin treated groups (n=4, each).

  1. Where are the controls? Since DMSO in high concentration is toxic and in low concentration it may have anti-inflammatory properties, it is important that you consider this and clearly state which concentration is used and its controls (item 2.1).
    ----- The reviewer’s comments were revised and improved as follows.

----- For examining in vivo effect of nobiletin, 5 µl of Nobiletin at 50 µM concentration were injected into the point 10 mm distal to the injury site immediately after SNI using micro-syringe. A vehicle solution (0.1% DMSO in 0.9% normal saline) was injected at the same location as the nobiletin treated group.

----- Twelve hours after DRG neuron culture, DMEM was changed and then nobiletin was treated into each well at a concentration of 0, 1, 30, 50, 100 or 200 µM. A vehicle solution (0.1% DMSO+0.1% EtOH in DMEM) was treated in culture dish of control group. Neurons were cultured for 36 hours and harvested for immunofluorescence staining. For detection of regeneration-related proteins, Schwann cells (5x106) were cultured onto 60 mm culture dish, and 12 hours later nobiletin was treated into each well at a concentration of 50 µM. Schwann cells were incubated for 72 h and then harvested for western blot analysis.

  1. What is the concentration of Nobiletin used in vitro? Was nobiletin diluted in DMSO? What is the concentration of DMSO? If so, the control must have the same DMSO concentration (item 2.2).

    ----- The reviewer’s comments were revised and improved as follows.
    ----- Twelve hours after DRG neuron culture, DMEM was changed and then nobiletin was treated into each well at a concentration of 0, 1, 30, 50, 100 or 200 µM. A vehicle solution (0.1% DMSO+0.1% EtOH in DMEM) was treated in culture dish of control group. Neurons were cultured for 36 hours and harvested for immunofluorescence staining. For detection of regeneration-related proteins, Schwann cells (5x106) were cultured onto 60 mm culture dish, and 12 hours later nobiletin was treated into each well at a concentration of 50 µM. Schwann cells were incubated for 72 h and then harvested for western blot analysis.

  2. Why did not you use one of the primary cultures (from uninjured tissue) to analyze the cytotoxicity of nobiletin? (item 2.2).

----- As pointed out by the reviewer, SH-SY5Y cells are a human derived cell used in vitro models of neuronal function and differentiation.

----- Regeneration of the sciatic nerve is closely related to the functional improvement of Schwann cells at the distal region of the injured nerves as well as the anterograde transport of regenerative proteins sent from motor neuron in the ventral horn of the spinal cord to the injury nerve.

----- In the present study, we want to know the toxicity of nobiletin in motor neurons, but it is known that the culture of primary motor neuron in the spinal cord is impossible.

----- For this reason, we investigated the toxicity of nobiletin using SH-SY5Y cells.

----- Thank you for your comment.

  1. It is known that nerve and DRG have connective tissue envelopes. What was the procedure to ensure sufficient purity of the primary cultures ? (item 2.3).

    ------ The reviewer’s comments were revised and improved as follows.
    ------ A dorsal root ganglion (DRG) is a cluster of neurons in a dorsal root of the spinal nerve, and sensory neurons of the sciatic nerve were normally located in DRG at lumbar 4-5 (Lyu et al., 2020). In our previous studies, we confirmed retrograde tracing of L4-5 DRG neurons after injection of DiI into the distal region to the damage site after SNI (Seo et al., 2009). Primary DRG sensory neurons and Schwann cells were prepared from L4-5 DRG and sciatic nerve, respectively, as described previously (Han et al., 2007; Seo et al., 2006, 2009).

  2. How much protein was used in electrophoresis? The WB was performed in how many cultures? What is “N” in vitro and in vivo? Did you use tissue pool in the electrophoseris? (item 2.4).

    ----- The quantitative analysis method requested by the reviewer was revised and improved as follows.

----- For in vitro experiment of DRG sensory neuron, a total 6 SD rats were used and this experiment was repeated 3 times with 2 rats each. To observe GAP-43 expression levels in primary cultured Schwann cells, SD rat (n=18) were assigned to normal control, vehicle and nobiletin treated groups (n=2, each) and it repeated 3 times.

----- For detection of regenerating axons in the distal region of the injured sciatic nerves, SD rat (n=10) were assigned to normal control (n=2), vehicle and nobiletin treated groups (n=4, each).

Discussion:

“Zhang et al. [21] also noted that nobiletin inhibited proliferation and differentiation of tumor cells such as neuroblastoma, breast cancer and fibrosarcoma [22-24]”.

Where does this argument fit into the results?

----- According to the reviewer’s comment, we replaced ‘paper written by Zhang et al., (2014)’ with ‘previous study written by Cho et al., (2015)’

Round 2

Reviewer 2 Report

Overall, in the revised manuscript, the author only replied parts of the concern raised by reviewer. Some missing pieces exist according to the current result. The specific comments are as follows:

  1. (Previous reviewer point 2) The author still cannot provide evidence whether the locally injected Nobiletin affected DRG in vivo. What is the rationale to inject Nobiletin in the injured nerve, instead of systemically given?
  2. (Previous reviewer point 3) The author still cannot provide longterm functional result. The reply was not directly to the point.
  3. (Previous reviewer point 4) Since the author agreed that 2 week is a proper period to demonstrate nerve regeneration after crush injury model, what was the  expression of BDNF, ERK/1/2 and AKT signal pathway at 2 weeks in figure 4?

Author Response

Thank you for your comments.
